# Detection of Carrageenan in Meat Products Using Lectin Histochemistry

**DOI:** 10.3390/foods10040764

**Published:** 2021-04-03

**Authors:** Marie Bartlová, Bohuslava Tremlová, Slavomír Marcinčák, Matej Pospiech

**Affiliations:** 1Faculty of Veterinary Hygiene and Ecology, University of Veterinary and Pharmaceutical Sciences Brno, Palackého tr. 1946/1, 61242 Brno, Czech Republic; bartlovam@vfu.cz (M.B.); tremlovab@vfu.cz (B.T.); 2Department of Food Hygiene and Technology, University of Veterinary Medicine and Pharmacy in Košice, Komenského 73, 04181 Košice, Slovakia; slavomir.marcincak@uvlf.sk

**Keywords:** agglutinins, food additives, galactose, hydrocolloids, light microscopy, polysaccharides

## Abstract

Carrageenan is a polysaccharide that is widely used in the food industry. Due to its water holding capacity, there is a higher risk of adulteration for economic reasons related to it. A verifiable method for detecting carrageenan is still missing in the food inspection sector. The detection of carrageenan in meat products is not well described. Our study describes lectin histochemistry as a novel approach for carrageenan detection. Within this study, the detection of carrageenan in meat products by lectin histochemistry is validated. Lectins of *Arachis hypogaea* (PNA) and *Bandeiraea simlicifolia* (BSA), specific for galactose units of carrageenan, were used. The samples included model meat products (ground chicken-meat products) and meat products from retail markets (chicken and pork hams, sausages, salami, and dried sausages). The limit of determination (LoD) of this method was set at 0.01 g kg^−1^. The method sensitivity for lectin PNA reached 1, and, for lectin BSA, it reached 0.96. Method specificity for lectin PNA was 1, and, for lectin BSA, it was 1.33. Cross-reactivity with other hydrocolloids tested was not confirmed. The results confirm that lectin histochemistry is suitable for detecting carrageenan in meat products.

## 1. Introduction

Carrageenan is a polysaccharide that is derived from red algae (Rhodophyceae) [1,2,3]. Its chains consist of units of D-galactose and 3,6-anhydro-glactose. These units are linked by α-1,3 and β-1,4 glycosidic bonds. Sulfate groups, which influence the resulting properties of the gel, are also included in their structure. Sulfate group content decreases solubility temperature as well as gel strength. Based on unit number and position, carrageenan can be divided into several groups. In its structure, carrageenan also contains polysaccharide residues such as glucose, xylose, uronic acids, methyl ethers, and pyruvate groups. In the food industry, the most commonly used carrageenan is kappa (κ), iota (ι), and lambda (λ) [1,3,4,5,6]. κ-carrageenan forms a solid and brittle gel; ι-carrageenan forms a flexible gel that is less strong than the gel formed by κ-carrageenan. λ-carrageenan does not form any gel but only viscous solutions. It is, therefore, used as a thickener. The advantage of this type of carrageenan is that it retains moisture in the product. κ- and ι-carrageenan are used in meat products, where they increase viscosity and water-binding capacity in the product [7,8,9]. The viscosity of the gel can be influenced not only by the type of carrageenan but also by its concentration, molecular weight, and, last but not least, temperature [6]. The molecular weight of commercially available carrageenan ranges from 100 to 1000 kDa [3,4]. Carrageenan contains negatively charged sulfate groups that form a strong bond with amine groups of proteins, especially with casein [10].

Carrageenan is used in the food industry not only for its gel-forming properties but also for its stabilizing, emulsifying, and thickening properties [1]. Usually, their content in food, in general, is 0.01–1.00 percent [11]. In compliance with European legislation [12], carrageenan is labeled with code “E407” for their refined forms and with code “E407a” for the unrefined form, often referred to as processed eucheuma seaweed. Except for infant formulas and follow-on formulas, there are no limits for this additive. Although carrageenan is an approved food additive, some studies suggest possible adverse effects on human health [13,14].

Analysis of carbohydrate polymers is not easy. That is because these additives can be added to the food in low concentrations or used in combination with other similar hydrocolloids for synergistic effect; there are components in the food matrix that may cause interference in the method used. Therefore, there is no general procedure for their analysis in food yet [15]. As reported in the literature, carrageenan can be determined by colorimetric methods based on the cationic dye carrageenan complex, which is then measured by an optical system (e.g., spectrofluorometer, spectrophotometer) [4,11]. In jelly, carrageenan can be detected by photometric titration [16]. Reverse-phase HPLC was developed by Quemener et al. [15] for custard powder, yogurts, and liver pâté. The potentiometric titration method was developed for cake jelly, ice cream, caramel, or salad dressing [17]. The DNA biosensor detection method was developed for carrageenan detection in pineapple jelly [11]. Carrageenan can also be detected nonspecifically using microscopic methods [18]. Bednářová et al. [19] detected carrageenan using light and electron microscopy in ham samples.

Lectins are proteins that specifically bind saccharides; isolated ones come from various sources, most often from plants, animals, fungi, and bacteria [20,21,22]. The principle of lectin histochemistry lies in lectin binding to a specific saccharide. The lectin–saccharide bond is visualized most often by chromogen, followed by microscopic determination. There are several methods in lectin histochemistry. The most common is using biotinylated lectin. The principle of this method is based on signal amplifying by binding avidin to biotinylated lectin. This complex is visualized by a horseradish peroxidase enzyme, labeled as avidin or streptavidin [20]. It is used for monitoring pathological and physiological processes on cell surfaces [23,24] and is considered a sensitive method capable of detecting various glycoconjugates.

Lectin histochemistry has not been used in food analysis yet. It may be difficult to detect carrageenan in meat products due to the strong bond between carrageenan and proteins. Meat products are a food commodity where carrageenan is used often. Carrageenan can be used as a means of food adulteration.

The aim of this work is to develop new detection methods for carrageenan detection in meat products with galactose-specific lectins. The developed methods were validated with qualitative method criteria.

## 2. Materials and Methods

### 2.1. Preparation of Model Samples

Model samples of a ground meat product with different polysaccharide additions were prepared. The model samples were produced from chicken breast muscle (from a trustful butcher), to which carrageenan was added in the following concentrations: 0.01, 0.1, 1, 10, and 100 g kg^−1^. These concentrations of carrageenan were selected to imitate smaller as well as higher concentrations of carrageenan in marked food. Other hydrocolloids were used in a concentration of 10 g kg^−1^. Moreover, 1 g of polyphosphates, 2.5 g of salt, and 10 mL of water were added to obtain a total of 100 g of the mixture. The mixture was homogenized in Vortex Thermomix (Vorwerk, Wuppertal, Germany) for 2 min. This was followed by cooking in an electric pot (Bielmeier Hausgeräte GmbH, Prackenbach, Germany), with the product core temperature at 70 °C for 10 min.

In model samples for cross-reactivity, the following concentrations were used: 3, 6, 9, and 12 g kg^−1^. Standard carrageenan κ (Sigma-Aldrich, St.Louis, MO, USA), carrageenan κ, ι and λ (Eurogum, Herlev, Denmark), ι refined and unrefined (Kerry, Tralee, Ireland), and starches, hydrocolloids (Raps, GmbH & Co. KG, Kulmbach, Germany), and spice (Vitana, Byšice, Czech Republic) were utilized.

The model samples were cut into pieces of 1 cm^3^, and 4 pieces were fixed with a 10% formaldehyde solution for at least 24 h. After fixation, the samples were dehydrated by an ascending alcohol series in a tissue processor (TP 1020, Leica, Wetzlar, Germany).

After dehydration, the samples were paraffin-embedded (Leica-Paraplast Plus, Leica Mikrosysteme Vertrieb, Wetzlar, Germany). Four paraffin blocks were prepared from each sample. The samples were cut using a rotating microtome (RM2255, Leica-Paraplast Plus, Leica Mikrosysteme Vertrieb, Wetzlar, Germany) to 5-µm thick sections on SuperFrost^®^ Plus glass (Thermo Scientific, Waltham, MA, USA). Four sections were made from each block, i.e., a total of 16 sections was prepared from each model sample. The sections were then dried and placed in a thermostat.

### 2.2. Preparation of Samples from the Market Network

Thirty-eight meat products were purchased from a retail market. Twenty samples declared carrageenan or processed eucheuma seaweed as contained ingredients. Eighteen samples did not report carrageenan or processed eucheuma seaweed in their formulation. These were mainly samples of ham, salami, sausages, and dried sausages. The number of meat products in each group reflects the frequency of carrageenan use in them. It included mainly samples of ham (*n* = 21), followed by sausages (*n* = 8), dried sausages (*n* = 7), and salami (*n* = 2) from pork or chicken meat. The products were selected randomly from Czech shops. The market samples were prepared in the same way as the model samples.

### 2.3. Buffer Preparation

The methodology of lectin histochemistry uses citrate buffer, lectin buffer, and Tris-buffered saline (TBS). Buffers were mixed according to the procedure by Brooks et al. [20]. B-Calleja solution was used for background staining [25].

Citrate buffer, pH 6

Citric acid monohydrate (2.1 g) was dissolved in 1 L of distilled water. The pH was adjusted to pH 6 using 2N NaOH.

Lectin buffer, pH 7.6

First, 60.57 g of Tris base, 87.0 g of sodium chloride, 2.03 g of magnesium chloride, and 1.11 g of calcium chloride were dissolved in 1 L of distilled water. Then, the pH was adjusted to the final pH of 7.6 using concentrated hydrochloric acid. Before use, the buffer was diluted 10 times in distilled water.

B—Calleja solution:

First, 1.0 mL of distilled water, 1.0 g of indigo carmine, and 200 mL of picric acid were mixed. Then, the solution was filtered before use.

### 2.4. Lectin Histochemistry

The method of lectin histochemistry is based on the method by Brooks et al. [20]. The method was optimized for the food matrix and the lectins used. It is formed with the following steps: dewaxing in xylene 15 min, twice; hydration in ethyl alcohol (2 × 10 min, 100% eth; 2 × 7 min, 96% eth.; 7 min, 70% eth.); 7 min washing in distilled water; 5 min, microwave carbohydrate retrieval in citrate buffer; 20 min cooling in room temperate; washing in lectin buffer twice; 20 min endogenous peroxidase blocking by hydrogen peroxide 3% in methanol; 5 min washing in lectin buffer twice; 60 min incubation by biotinylated lectin (*Arachis hypogaea* (PNA) or *Bandeiraea simlicifolia* (BSA; Sigma-Aldrich s. r. o., St.Louis, MO, USA), diluting 1 µL in 1 mL; 5 min washing in lectin buffer three times; 30 min incubating in reagent A and reagent B ABC (Vector Laboratories, Inc.; Burlingame, CA, USA); 5 min washing in lectin buffer, three times; 5 min visualization by DAB substrate kit, (Vector Laboratories, Inc.; Burlingame, CA, USA); 5 min washing in distilled water; 5 min background staining in B-Calleja; 5 min washing in distilled water; 7 min bath in ethanol 90%; 7 min bath in ethanol 100%; 7 min bath in xylene p.a. twice; solacryl mounting. The samples were examined with a Nicon Eclipse Ci light microscope (Nikon, Minato, Japan).

### 2.5. Statistical Processing

The results were statistically processed in Unistat 6.0 (Unistat Ltd., London, UK). McNemar’s test on a contingency table was used to compare lectin reactivity according to Pospiech et al. [25]. Multiple comparisons with a *t*-distribution test were used to compare the marketed product results and the type of carrageenan used (E407 and E407a).

### 2.6. Specificity and Sensitivity

The sensitivity and specificity of the method were determined from the results of cross-reactivity for model products. The sensitivity of the test is expressed by the ratio of positive and truly positive samples. Specificity is the ratio of negative and truly negative samples. It was calculated according to Trullols et al. [26].

## 3. Results and Discussion

### 3.1. Determination of Lectin Concentration

The reactivity of PNA and BSA lectins was verified at various concentrations to determine the optimal lectin concentration for the lectin-saccharide reaction. Biotinylated lectin PNA is specific to the galactosyl (β-1,3) N-acetylgalactosamine. Biotinylated lectin BSA has a major affinity for terminal α-d-galactosyl residues, with a secondary affinity for terminal N-acetyl-α-d-galactosaminyl residues (Sigma-Aldrich, St.Louis, MO, USA).

A lectin dilution of 10 through 0.1 μg mL^−1^ was used. The lectin was diluted in lectin buffer. A very strong signal of carrageenan was found in both evaluated lectins, even at a dilution of 0.1 μg mL^−1^ (Table 1).

As reported by Brooks et al. [20], the optimal concentration for most lectins is 10 μg mL^−1^. However, there are exceptions. For example, optimal lectin concentrations from the plant *Phytolacca americana* is 1 μg mL^−1^. Conversely, the recommended amount of lectin from the plant Limulus polyphemus is 100 μg mL^−1^. The optimal concentration of lectins for the analyses within this work was set as 1 μg mL^−1^. The intensity of staining at this concentration was very strong for both lectins (Table 1). The selected concentration is also consistent with antibody dilution in immunohistochemical methods [27]. In the case of higher dilutions, the risk of a weaker lectin response (staining intensity) is increased, especially in the case of high levels of carrageenan in the matrix. From immunohistochemical methods, it is also known that high antibody concentration does not always mean better signal quality [28]. However, the benefit of higher dilutions is the reduced cost of the analysis.

In addition to lectin concentration, lectin reactivity may be affected by fixative solutions. For example, the sensitivity of *Griffonia simplicifolia* lectin may be reduced if the tissue is fixed with formaldehyde. Conversely, the sensitivity of this lectin may increase if the sample is fixed with ethanol and acetic acid [29]. The bond between lectin and saccharide may also be impaired due to the mounting medium—paraffin. Paraffin can cause protein denaturation. However, this can be avoided by dewaxing using xylene [29], as was the case with the method used in this study.

### 3.2. Method Repeatability

Method repeatability is an indicator of method accuracy. Four different samples were examined; each sample was examined at 20 sections. Specifically, these were a model sample with κ-carrageenan (concentration 10 g kg^−1^), a model sample with ι- carrageenan (concentration 10 g kg^−1^), a model sample with λ- carrageenan (concentration 10 g kg^−1^), and a randomly chosen positive sample from the market network, M21 (sample of meat product from the market network). The evaluation was carried out by three trained evaluators—a histology laboratory assistant, a doctor of veterinary medicine, and a student. The results are presented in Table 2. In some cases, the histological tissue was lost, and, therefore, the number of sections examined varied between lectins and evaluators. This phenomenon is also common in other histological methods [27,30,31]. Frequent reasons are poor section fixation, pH value of food matrix, long-time sample preparation, and inadequate antigen retreatment.

The repeatability of the method was 100% in all cases. Carrageenan was detected in all selected matrices (model samples and the market meat product; Figure 1) and in all repetitions. The number of positive sections may vary depending on the quality of the individual sections and the loss of histological tissue. Loss of histological tissue is easily recognized by the evaluator and does not result in a false interpretation of the results. In one case, a section was dubious. This interpretation was caused by a nonspecific bond or contamination during processing in laboratory work.

### 3.3. Limit of Determination (LoD)

In this article, a qualitative method of lectin histochemistry is used. Thus, the LoD parameter was utilized to evaluate this method. The LoD is an important parameter of all analytical methods [26]. In this work, carrageenan LoD was determined in the model samples of a meat product. For histology methods, LoD was determined only qualitatively by a trained evaluator (Table 3).

The results clearly show that when using the lectin histochemistry method, carrageenan was detected at concentrations as low as 0.01 g kg^−1^ for all three types of carrageenan (κ, ι, and λ) for both the PNA and BSA lectins tested. Samples with lower carrageenan concentration were not included in the analysis due to the impossibility of homogenous processing of the model samples. A much lower LoD was measured by Hassan et al. [11], who detected carrageenan using a DNA sensor; the LoD of that method was 0.08 mg L^−1^; however, that method is an indirect one. In contrast to Hassan et al. [11], lectin histochemistry is a direct method that directly demonstrates carrageenan macromolecules. In their work, Ziółkowska et al. [16] focused on carrageenan detection by photometric titration, where the LoD for methylene blue equaled 1.6 mg L^−1^ and, for toluidine blue, 2 mg L^−1^. Hassan et al. [17] detected carrageenan by potentiometric titration using polyion sensors. The LoD values vary depending on the matrix contained in the sensor. The LoD values ranged from 0.05 to 2.81 µg mL^−1^. Conversely, greater LoD was observed compared to lectin histochemistry in Ling and Heng [32], who determined the carrageenan–methylene blue complex using an optical sensor in their work. The LoD of that method was found to be 80 mg L^−1^. In comparison to the LoD values of some of these methods, lectin histochemistry does not have the lowest LoD value. The LoD value is also influenced by the type of matrix analyzed and its consistency. The advantage of lectin histochemistry is its ability to detect carrageenan in differently formed matrices. With regard to the variability of meat products, we consider this universal applicability as one of the important advantages of lectin histochemistry.

### 3.4. Cross-Reactivity

For immunochemical methods, cross-reactivity with another protein is one of the most common causes of false-positive reactions in foodstuffs [33]. In the case of lectin histochemistry, cross-reactivity with other saccharides can be assumed to be similar to that demonstrated in human serum in lactose-specific lectins [34] and in fish [35]. In foodstuffs, cross-reactivity has not yet been verified. Therefore, a wide range of potentially or widely used polysaccharides in food and raw materials with higher polysaccharide content was verified. Regarding the possible polar bond between lectin and protein, protein raw materials were also tested. The samples used were predominantly model samples of meat products with additional ingredients or raw materials (Table 4).

The results showed high specificity of lectin-polysaccharide binding. Only 5.88 percent of the samples was false-positive for PNA and BSA. The most commonly used polysaccharide in the meat industry is starch. Both lectins do not bind to starch because they contain glucose monomeric units. Corn (CR3, CR4), lupine (CR5), and amaranth (CR6) contain a considerable amount of starch and are, therefore, mainly composed of glucose. Fibre (CR7), whey protein (CR8), whey powder (CR10), emulac cc (CR9), and pea protein (CR13) are also free from galactose in their structure, and the reactivity of lectins with them has not been confirmed. Similarly, the guar gum (CR15) polysaccharide does not bind to lectins because monomer units are galactomannan. Although the core is galactose, mannose is attached to the side chains to prevent lectin binding [36]. Arabic gum (CR19) core is formed from monomeric units of galactoses, to which arabinose, rhamnopyranose, and glucuronic acid units are bound; therefore, the signal is negative [37]. Cross-reactivity was also not confirmed for the xanthan gum (CR34) sample because the backbone of xanthan gum is composed of glucose, to which glucuronic acid and mannose are attached [38].

The sample with the addition of agar-agar (CR14) was evaluated as dubious. Agar-agar contains D- and L-galactose in its structure [39]. Thus, the lectin was probably bound to D-galactose but did not bind to L-galactose. Cross-reactivity was not confirmed with tragacanth gum (CR35) because its core consists of arabinose, xylose, and galacturonic acid [40]. The model samples with the additions of alginate (CR16) and carboxymethylcellulose (CR17) were evaluated as dubious. In some parts of the preparations, the signal intensity was weak; in other parts, the fragments remained completely unstained. The Alginate core consists of mannuronic acid and guluronic acid [41]. Carboxymethylcellulose is a cellulose derivative. It is a polysaccharide, the structure of which consists of glucose units [42,43]. In the case of alginate, weak signal intensity in some sections was a result of cross-linked activity, which was confirmed with alginans and a variety of substances commonly used in pharmacies to regulate drug release [44]. A similar effect was confirmed with methyl cellulose, hydroxymethyl cellulose, and carboxymethyl cellulose [45]. Carob (CR18) is a brown pod with a wrinkled surface and very hard seeds inside [46]. The brown color in the histology section is similar to the brown color of the chromogen used, which is a reason for the false-positive results. The morphological structures are shown in Figure 2.

Sample CR11 was evaluated to be false-positive. The reason for false-positivity was the presence of caraway. The misclassification was due to the natural brown pigmentation of caraway seeds [47]. This reason was confirmed by positive results for samples CR30, CR31, and CR32, which were model samples of meat products with the addition of 1% caraway and pepper. These spices can be distinguished from carrageenan by the characteristic of their typical morphological structure (Figure 3); however, in the case of the caraway layer or the sclerenchymatic cell layer fragments in pepper, it cannot be clearly distinguished from a small fragment of carrageenan. Another chromogen that does not correspond to natural pigmentation by its color could be used in order to differentiate the spices. In terms of the paraffin block technique with synthetic resin mounting, for example, green-colored HistoGreen can be used [48]. However, caraway contains a small amount of galactose in its structure [49]; therefore, a positive lectin–galactose reaction cannot be excluded even if another chromogen is used.

According to Trullols et al. [26], it is appropriate to use real samples for the validation of qualitative methods in order to validate the influence of the matrix and the method of processing on the results. For this purpose, model products manufactured according to common standards for meat products were evaluated. Specifically, the matrices of ham and poultry sausages were verified. Reactivity of the compared lectins in the real model products is shown in Table 5.

The model ham sample (CR-MP-3) injected with 20% nitrite salt mixture 2.5%, without carrageenan addition, was false-positive for BSA lectin, although for lectin PNA, it was in accordance with the provided declaration. The opposite was the case with sample CR-MP-8, the model sample with the addition of carrageenan 0.3%. The BSA reactivity result was false-negative. Similarly, for sample CR-MP-19, which contained pure standard carrageenan (Sigma-Aldrich, USA), the lectin–polysaccharide reaction was false-negative. Different reactivity of different lectins with the same saccharide specificity was also described [20]. Another reason for false-negative results may be the low carrageenan concentration in the product, where the sample selected contains carrageenan-free tissue. The minimum number of samples to be evaluated may vary. There are insufficient studies for lectin histochemistry; however, this specific question was verified for the methods based on immunohistochemical techniques. As reported by Besusparis et al. [50], three histological sections are sufficient; nonetheless, Goethals et al. [51] asserted that 8–12 sections are suitable. Our applications on food matrices suggest that 8 sections should be examined [52]. Statistically significant differences between lectins alone or between lectins and declared content were not demonstrated (*p* = 1.00) by McNemar’s test. As sample CR-MP-19 shows, FN results may also occur at higher concentrations due to sampling. For qualitative methods, sensitivity and specificity tests are recommended. These tests show to what extent the diagnostic tests provide definitive information about the presence or absence of the target analyte [26,53]. Sensitivity is the ratio of positive and truly positive results—lectin PNA reached 1, while BSA reached 0.96. Specificity is the ratio of negative and truly negative samples. Method specificity was 1 for PNA and 1.33 for BSA.

### 3.5. Detection of Carrageenan in Meat Products from the Market Network

Interferences of the analyte or methods with the matrix should be tested on real samples. Thus, the determination of carrageenan by lectin histochemistry was tested on products purchased from the market network. The results are presented in Table 6.

A comparison of the results demonstrated that there was no statistically significant difference (*p* ˂ 0.05) between the tested PNA and BSA lectins and between the positive and negative sections. For lectin histochemistry, deviations in lectin reactivity with the declared ingredients in some products were found.

Samples No. M12, M18, M20, M26, M34 were evaluated as positive for both lectins, although the samples were declared negative. The most likely reason was the absence of carrageenan in the declared ingredients of the products. The positive reaction of both lectins on most histological sections shows an even representation; additionally, the morphological structure corresponds to carrageenan, which indicates an incorrect declaration. Other studies have also confirmed that there is a difference between the method result and the declaration when validating methods on commercial samples [54,55]. Sample No. M37 was declared free of carrageenan. Nevertheless, both lectins showed a positive response in some sections, namely, PNA (6) and BSA (4). The product contained a large amount of spices, which is probably the reason for the false-positive result. The false-positive result was also indicated by the number of false-positive sections as, when carrageenan is normally added to meat products, its distribution in the product is even and the positive result would be in all the sections analyzed. The false-positivity of spices, especially caraway, is related to its natural brown color [47] and its galactose content [49]. In addition to incorrect identification of carrageenan in the product due to identical pigmentation with chromogen, a false-positive reaction in lectin histochemistry may also be caused by nonspecific protein–protein binding due to the lack of recognition of the carbohydrate by lectin. The solution to this problem is to use a control reaction with unlabeled carbohydrates as competitive inhibitors [56].

Sample M23 was negative for both lectins, despite the fact that carrageenan content was stated in the declaration. Possible reasons were insufficient mixing of raw materials in the product, their absence in the analyzed sample, or incorrect declaration by the manufacturer. The issue of false-positive and false-negative results is also known in immunochemical and immunohistochemical methods [57,58,59,60]; this is, obviously, also a source of some uncertainty in lectin histochemistry [20,61].

The differences in immunoreactivity between processed eucheuma seaweed (E407a) and carrageenan (E407) were not confirmed (*p* < 0.05). The difference between carrageenan and processed eucheuma seaweed is in refining. Carrageenan in food is labeled as E407 and its semirefined form is labeled as E407a; they differ slightly in cellulose content [5]. According to Sedayu et al. [3], cellulose content, together with other plant residues, makes up about 20–30 percent of semirefined carrageenan.

The advantage of lectin histochemistry, compared to nongalactose-based detection methods, is that it is a direct method with a high specificity of lectin carbohydrate-binding that provides accurate information about the sample’s structure. The incorporation of carrageenan into the matrix was clearly visible in all positive samples. Image analysis can also distinguish the type of carrageenan based on the color intensity of the precipitate; its typical shade is measured in the RGB color space [62]. Another benefit of this method is that it can be applied to a small sample (1 cm^3^), although a small amount of sample can lead to a false-negative result (e.g., Sample M23). In contrast, there are methods that require using large amounts of samples. For example, Hassan et al. [11] stated that some methods based on the gravimetric determination of sulfates require large amounts of samples to obtain sufficient sulfate for analysis. The results of this research may be a new tool that can be used in the food inspection system. A disadvantage of lectin histochemistry involves potential false-positive reactions. This can be prevented by including positive and negative control samples into the analysis [20] and by training the evaluators.

Sample processing for lectin histochemistry is not complicated but can be time-consuming with respect to the type of matrix. However, this may also be a problem for other methods, where the sample must be subjected to extensive preparation and purification to remove components that could cause interference [11]. Quemmener et al. [15] detected carrageenan by HPLC, where the measurement is preceded by extraction or lyophilization of the sample and subsequent methanolysis to release 3,6-anhydrogalactose, which is subsequently determined. The authors pointed out that the advantage of this method is that there is no need to remove lipids and proteins from the sample. In contrast, the method of determining carrageenan using a DNA biosensor is not demanding for sample preparation [11]. However, the method was applied to a pineapple jelly matrix that did not contain lipids or proteins. On a similar matrix—jelly sachet—carrageenan was detected by Ziółkowska et al. [16]; the advantage of colorimetric methods is the undemanding preparation process.

## 4. Conclusions

The lectin histochemistry method was validated for the detection of κ-, ι-, and λ-carrageenan in meat products. The LoD was set at 0.01 g kg^−1^; an economically suitable concentration of lectins was determined to be 1 ug mL^−1^. The reactivity of PNA and BSA lectins was verified, and there was no statistically significant difference in the reactivity between them (*p* > 0.05). The lectin histochemistry method is independent of the evaluator and laboratory conditions (repeatability and reproducibility were 100 percent). The suitability of the method for a meat product matrix was verified by a sensitivity and specificity test. Method sensitivity was 1 for PNA and 0.96 for BSA. Method specificity was 1 for PNA and 1.33 for BSA. Cross-reactivity was demonstrated for some components; alginate and carboxymethyl cellulose samples exhibit nonspecific cross-linking with lectin. When spices are present, misinterpretation of the result may be due to the similar pigmentation of some spices and the chromogen precipitate. However, a trained evaluator can eliminate these errors based on the morphological structure. Validation of lectin histochemistry on products from the market network confirmed the suitability of the method for the analysis of real samples. Our results also show that carrageenan may be used in meat products and not included in the product ingredient list.

## Figures and Tables

**Figure 1 foods-10-00764-f001:**
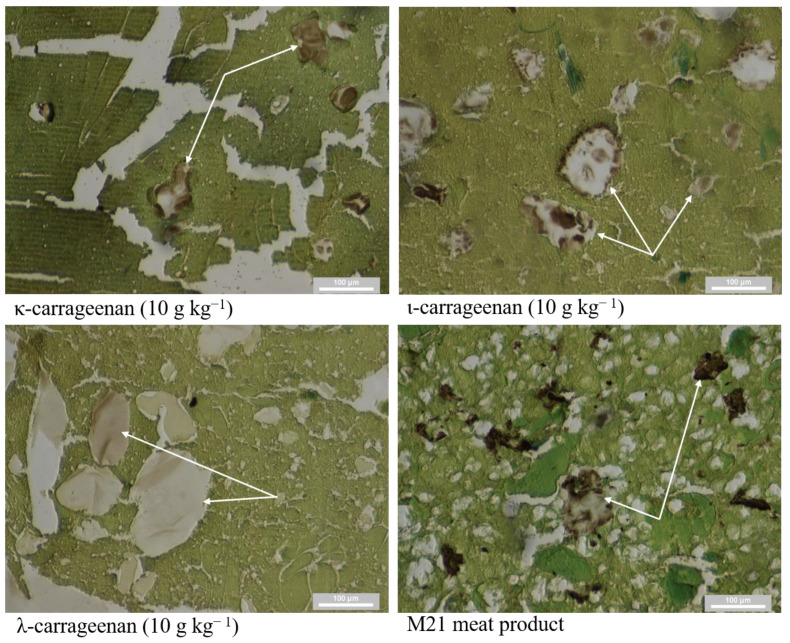
Positive reactions in tested matrices: carrageenan visualized in brown (arrows) model samples (κ-, ι- and λ-carrageenan, with a concentration of 10g kg^−1^) and a marketed meat product (M21).

**Figure 2 foods-10-00764-f002:**
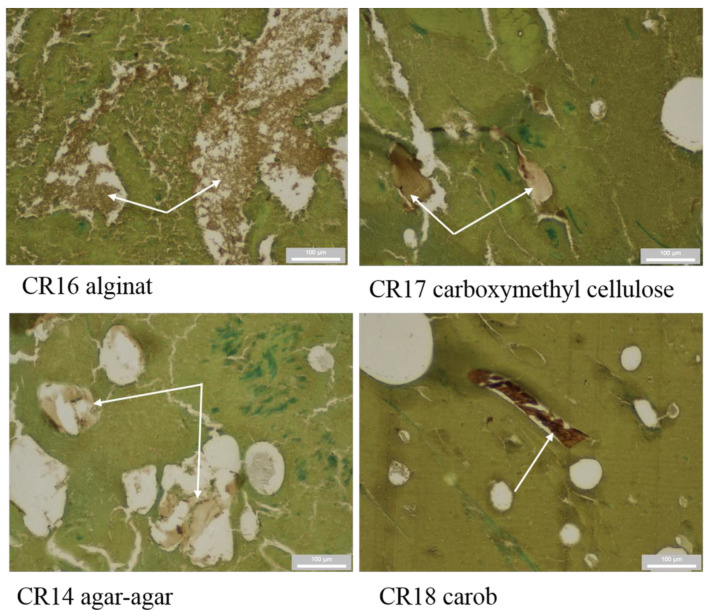
Structure of alginate, carboxymethyl cellulose, agar-agar, and carob (arrows).

**Figure 3 foods-10-00764-f003:**
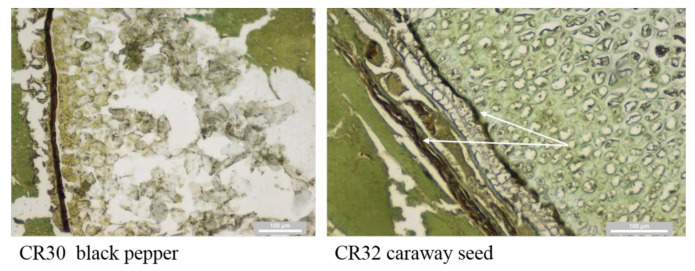
Structure of black pepper (CR30) and caraway (CR32) by lectin histochemistry; natural brown pigmentation.

**Table 1 foods-10-00764-t001:** Signal intensity of model carrageenan sample using the lectin concentration sequence.

Concentration (μgmL^−1^)	Dilution	Lectin PNA	Lectin BSA
10.00	1:100	+++	+++
1.00	1:1000	+++	+++
0.40	1:2500	+++	+++
0.20	1:5000	+++	+++
0.13	1:7500	+++	+++
0.10	1:10,000	+++	+++

Note: Signal intensity is from + (weak) to +++ (very strong). PNA—lectin *Arachis hypogaea*; BSA—lectin *Bandeiraea simlicifolia.*

**Table 2 foods-10-00764-t002:** Results of method repeatability.

Sample	Evaluators					
	1		2		3	
	Lectin PNA	Lectin BSA	Lectin PNA	Lectin BSA	Lectin PNA	Lectin BSA
κ-carrageenan (10g kg^−1^)	20/0	17/0	20/0	16/0	20/0	16/0
ι-carrageenan (10g kg^−1^)	20/0	20/0	20/0	20/0	20/0	20/0
λ-carrageenan (10g kg^−1^)	20/0	20/0	19/±	20/0	20/0	20/0
M21	20/0	19/0	20/0	19/0	20/0	19/0

Note: number of positive sections/number of negative sections; ± dubious; PNA—lectin *Arachis hypogaea*; BSA—lectin *Bandeiraea simlicifolia.*

**Table 3 foods-10-00764-t003:** Results of LoD determination of κ-, ι- and λ-carrageenan in model samples of meat products.

Carrageenan Concentration [g kg^−1^]	Lectin PNA			Lectin BSA		
	κ	ι	λ	κ	ι	λ
0.01	P	P	P	P	P	P
0.1	P	P	P	P	P	P
1	P	P	P	P	P	P
10	P	P	P	P	P	P
100	P	P	P	P	P	P

Note: P—positive; PNA—lectin *Arachis hypogaea*; BSA—lectin *Bandeiraea simlicifolia*; κ-carrageenan, ι-carrageenan, λ-carrageenan.

**Table 4 foods-10-00764-t004:** Results of cross-reactivity for raw materials.

Identification or Number of Samples	Composition of Model Sample	Lectin PNA	Lectin BSA
CR1	native potato starch	N	N
CR2	native tapioca starch	N	N
CR3	native corn starch	N	N
CR4	waxy corn	N	N
CR5	meat + 1% lupine	N	N
CR6	meat + 1% amaranth	N	N
CR7	meat + fiber	N	N
CR8	meat + 1% wheat protein	N	N
CR9	meat + emulac cc	N	N
CR10	meat + whey powder	N	N
CR11	meat + tonsils + brain + caraway	FP	FP
CR12	meat + 1% pea flour	N	N
CR13	meat + 1% pea protein 80	N	N
CR14	meat + 1% agar-agar	±	±
CR15	meat + 1% guar gum	N	N
CR16	meat + 1% sodium alginate	±	±
CR17	meat + 1% carboxymethylcellulose	±	±
CR18	meat + 1% carob	FP	FP
CR19	meat + 1% gum arabic	N	N
CR20	meat + 0.9% E407 + 0.3% polyphosphate	P	P
CR21	meat + 0.9% E407 + 2.1% nitrite salt	P	P
CR22	meat + 0.9% E407 + 0.05% ascorbic acid	P	P
CR23	meat + 0.9% E407 + 2.1% table salt	P	P
CR24	meat + 0.9% E407 + 1% emulac cc	N	±
CR25	meat + 0.9% E407 + 5% collagen	N	N
CR26	meat + 0.9% E407 + 1% soy	P	P
CR27	meat + 0.9% E407 + 1% whey	P	P
CR28	meat + 0.9% E407 + 0.3% polyphosphates + 2.1% nitrite salt	P	P
CR29	meat + 0.9% E407 + 0.3% polyphosphates + 2.1% nitrite salt + 0.05% ascorbic acid	P	P
CR30	meat + 1% black pepper crushed	FP	FP
CR31	meat + 1% white pepper ground	FP	FP
CR32	meat + 1% caraway ground	FP	FP
CR33	meat + 1% konjac gum	N	N
CR34	meat + 1% xanthan gum	N	N
CR35	tragacanth gum	N	N

Note: P—positive; N—negative; FP—false-positive; ± dubious; PNA—lectin *Arachis hypogaea*; BSA—lectin *Bandeiraea simplicifolia.*

**Table 5 foods-10-00764-t005:** Results of cross-reactivity for model products.

Sample	Lectin PNA	Lectin BSA	Declaration	Notes
CR-MP-1	N	N	N	pâté without E407
CR-MP-2	P	P	P	pâté with E407
CR-MP-3	N	FP	N	ham—injection 20% nitrite salt 2.5%, without carrageenan
CR-MP-4	P	P	P	ham—injection 20% nitrite salt 2.5%, κ- carrageenan 1%
CR-MP-5	P	P	P	ham—injection 20% nitrite salt 2.5%, λ-carrageenan 1%
CR-MP-6	P	P	P	ham—injection 20% nitrite salt 2.5%, ι-carrageenan 1%
CR-MP-7	N	N	N	MP—without additive
CR-MP-8	P	FN	P	MP—E407 0.3%
CR-MP-9	P	P	P	MP—E407 0.6%
CR-MP-10	P	P	P	MP—E407 0.9%
CR-MP-11	P	P	P	MP—E407 1.2%
CR-MP-12	P	P	P	MP—E407a 0.3%
CR-MP-13	P	P	P	MP—E407a 0.6%
CR-MP-14	P	P	P	MP—E407a 0.9%
CR-MP-15	P	P	P	MP—E407a 1.2%
CR-MP-16	P	P	P	MP—κ-carrageenan standard 0.3%
CR-MP-17	P	P	P	MP—κ-carrageenan standard 0.6%
CR-MP-18	P	P	P	MP—κ-carrageenan standard 0.9%
CR-MP-19	P	FN	P	MP—κ-carrageenan standard 1.2%
CR-MP-20	P	P	P	MP—λ-carrageenan 0.3%
CR-MP-21	P	P	P	MP—λ-carrageenan 0.6%
CR-MP-22	P	P	P	MP—λ-carrageenan 0.9%
CR-MP-23	P	P	P	MP—λ-carrageenan 1.2%
CR-MP-24	P	P	P	MP—κ-carrageenan 0.3%
CR-MP-25	P	P	P	MP—κ-carrageenan 0.6%
CR-MP-26	P	P	P	MP—κ-carrageenan 0.9%
CR-MP-27	P	P	P	MP—κ-carrageenan 1.2%

Note: P—positive; N—negative; FP—false-positive, FN—false-negative; MP—model sample; E407—carrageenan; E407a—eucheuma seaweed; PNA—lectin *Arachis hypogaea*; BSA—lectin *Bandeiraea simlicifolia.*

**Table 6 foods-10-00764-t006:** Results of lectin histochemistry in meat product samples from the market network.

Sample	Lectin PNA	Lectin BSA	Additive Used
	Positive/Negative *	Positive/Negative *	
M1	8/0	8/0	P (E407)
M2	8/0	8/0	P (E407a)
M3	6/2	5/3	P (E407)
M4	8/0	8/0	P (E407)
M5	8/0	7/1	P (E407)
M6	8/0	8/0	P (E407)
M7	8/0	8/0	P (E407a)
M8	8/0	8/0	P (E407)
M9	7/1	5/3	P (E407)
M10	8/0	7/1	P (E407)
M11	8/0	8/0	P (E407a)
M12	8/0	8/0	N
M13	8/0	5/3	P (E407)
M14	8/0	8/0	P (E407)
M15	7/0	8/0	P (E407)
M16	8/0	8/0	P (E407)
M17	2/6	2/6	N
M18	8/0	8/0	N
M19	6/0	8/0	P (E407)
M20	7/1	8/0	N
M21	8/0	8/0	P (E407)
M22	8/0	8/0	P (E407)
M23	0/7	0/7	P (E407)
M24	8/0	8/0	P (E407a)
M25	3/5	3/5	N
M26	8/0	5/3	N
M27	0/8	0/8	N
M28	0/8	0/7	N
M39	0/8	0/8	N
M30	0/8	0/8	N
M31	2/6	3/5	N
M32	2/6	2/4	N
M33	0/8	0/8	N
M34	5/3	5/1	N
M35	0/8	1/7	N
M36	0/7	0/5	N
M37	6/2	4/2	N
M38	0/7	0/5	N

Note: * Number of positive sections/number of negative sections; E407—carrageenan; E407a—processed eucheuma seaweed; N—negative (does not contain E407 or E407a); PNA—lectin *Arachis hypogaea*; BSA—lectin *Bandeiraea simlicifolia.*

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
