# Peer review of "Detection of Carrageenan in Meat Products Using Lectin Histochemistry"

_foods, 2021, doi:10.3390/foods10040764_

Round 1
Reviewer 1 Report
The manuscript represents an interesting study on the possibility of qualitative detection of carrageenan by the lectin histochemistry method in chicken products and the establishment of basic validation criteria for this food matrix.
I have provided my comments as follows.
- In the title of the manuscript, the authors should emphasize the type of meat that was the subject of the study.
- The abstract contains some unnecessary paragraphs that rather belong in the introductory chapter (especially the first three sentences of the abstract). At the same time, the abstract lacks important information about how the experiment was conducted. Moreover, it would be good to add in the abstract the practical application of the results.
- Considering the objective of this work, the authors should expand the chapter Introduction with information about the method of qualitative determination of carrageenan by the method of lectin histochemistry. The introduction itself contains very little information on the use of microscopic methods.
- The results of this study should be better described in the discussion section in the context of the published literature. Unfortunately, there is no discussion of the limitations of the study (the author's suggestion is to add what the limitations of the study are).
- Conclusions: Please describe the research limitations for future research and reiterate your main findings.
Author Response
Dear reviewer,
Thank you for these pertinent comments
We agree that some parts of our article were not sufficiently specific in terms of details provided. We revised the major objective in the abstract and Introduction. The title we prefer not more specific because it our study was used four different group of meat product and their description in title will complicate it.
In the introduction, we paid more attention to the specification basic information about microscopy method.
The Results and Discussion section was updated in harmony with your comments, mainly method limitation.
In all text, we also corrected minor mistakes. We provide also changes according to the other reviewer and some parts was changed according to these. We believe that all changes make manuscript more relevant and read able for readers.
In the Conclusion, we added future research prospects and we reformulated them to correspond to our results better.
We also performed professional editing to ensure that the text is well phrased and free from typographical and grammatical errors. All the changes are highlighted in the manuscript.
Reviewer 2 Report
Comments to Author:
The manuscript is about using lectin histochemistry to qualitatively detect carrageenan in processed meat products and to establish basic validation criteria for various meat product matrixes.
I found the paper at times very difficult to read (especially the introduction and materials and methods) with a lack of details. The authors have not made a satisfactory effort to motivate the novelty of the paper and how it advances the knowledge within this domain. The method has not been modified in any way (from what I understand at least from reading the text) and it feels somewhat archaic. The authors need to consider improving the language of the paper, preferably by having a native English speaker to revise the text. The data and analyses were not presented appropriately as they lack details. The authors need to improve this to enable the reproducibility of the method and results, but also to provide the reader with a better understanding of the samples they used and the reasoning for using the specific samples for the study. On a positive note, I find that the paper has a generally good results and discussion section. However, the authors need to make various improvements (as pointed out under Comments to consider).
Comments to consider:
Title (L2-3)
The paper has a suitable title.
Abstract (L11-21)
L11: Carrageenan is a… Please check the rest of the manuscript’s grammar.
The abstract is not well-written. It is unclear what the main aim of the research was as well as its significance. It needs to be improved.
The abstract should be one paragraph.
L18: Specify the type of meat products.
Keywords (L22-23)
Consider arranging alphabetically.
Introduction (L25-80)
L36: Not clear what product’s texture may be damaged.
L38-44: A lot of facts are given about carrageenan, but it is not clear to the reader by this is important. The authors need to reconsider the structure of the introduction with a “golden thread” from the start to the finish as the flow of the text is not good.
L45-46: This has been mentioned before.
L46-47: It is unclear in which products the content is 0.01-1.00%.
L47: Please mention which regulations: EU?
L56-L66: The text requires improvement and it not well written and difficult to read. It can be written more concisely.
L67-L69: What is the purpose of this paragraph?
L78: It is unclear how the authors came to this aim. They need to provide sufficient background to motivate the novelty and importance of the detection of carrageenan by lectin histochemistry. It is unclear why meat products were selected. The authors also need to state if this method has been used on meat products before, if so, what is then the significance of this study. If not, then they need to mention why it is important to detect carrageenan qualitatively in meat products (what does it mean for the industry?). As a reader, I am not convinced that this study has any novelty.
Materials and methods (L81-157)
L83: More details about the nature of the meat product are required. How was it sampled, stored, processed, etc?
L84: Why was chicken breast selected?
L85: Why did the authors use the specific concentration levels?
L89: How were the samples cooked? In a mould?
The lack of details is a major concern. The authors need to improve this.
L105: Again, a lack of detail in this section. What type of meat products were sampled from the market? The authors need to give an overview of all the collected samples, mentioning the number of samples per product category. Was there a strategy for market sampling?
L106-107: Write out 38, 20 and 18 in full as these numbers are at the start of the sentences.
Results and Discussion (L158-425)
L196-197: Be consistent with the use of symbols for kappa carrageenan, iota carrageenan and lambda carrageenan.
L224: LoD abbreviation should be after the first mention of “limit of the determination”.
How was the LoD determined? By the evaluators/examiners? It is unclear. Please explain.
L229: It is unclear how the reader should interpret Table 3. Please explain the symbols used in the table.
L270-299: It would be good for the authors to refer to the specific products (sample identification) in Table 4 that are being discussed.
Overall, the discussion of the results is good. The authors can just revise and improve the language in some sentences.
L292: Refrain from using the subjective “we”.
For all the figures, it is recommended to explain in the captions what the arrows are pointing at in the different images.
L371: It is an interesting result, which raises the question of whether mislabelling (incorrect declaration) of the ingredients took place or if the qualitative lectin histochemistry method is accurate enough. Did the authors consider some quantitative confirmatory techniques as well to verify these results? Could there be a matrix effect? It is very complicated to draw definitive conclusions without solid proof.
L402: Please specify the “other detection methods”.
L402: I would suggest making this a new section. Something along the line of “General considerations and future outlook”. Include some aspects of how it could be applied soon using, for example, the latest technological developments.
Conclusions (L426-442)
L441: How was it confirmed that “carrageenan is used in meat products and are not included in the product ingredients list”? As the authors did not execute any additional confirmatory analyses.
Author Response
Dear reviewer,
Thank you for these pertinent comments
We agree that some parts of our article were not sufficiently specific in terms of details provided. We revised the major objective in the Abstract, Introduction and Materials and Methods.
The keywords were corrected according to yours comments.
In the introduction, we paid more attention to the readability. We reworded parts of detection methods for carrageenan. We also restated the goal of article to be more specific.
The Materials and Methods section was updated in accordance with your comments, mainly description of samples.
The Results and Discussion section was updated in accordance with your comments, mainly symbol unification as well as Table and Figure description.
About your comment: we have considered some other methods for carrageenan determination. But unfortunately, detection methods, which can specifically detect carrageenan, are not developed. In present time it is not possible to provided other direct confirmatory methods for meat products. The cross-reactivity confirmation and retail market samples evaluation was the only way how to show and disscus our results. We will continue in this research and we hope that we will be able answer these questions later.
In the Conclusion, we mitigated the power of the claim.
In all text, we also corrected minor errors. We provided also changes according to the other reviewer and some parts were changed according to this. We believe that all changes make manuscript more relevant and improve readability for readers.
We also performed professional editing to ensure that the text is well phrased and free from typographical and grammatical errors. All the changes are highlighted in the manuscript.